# New Insight into the Potential Protective Function of Sulforaphene against ROS−Mediated Oxidative Stress Damage In Vitro and In Vivo

**DOI:** 10.3390/ijms241713129

**Published:** 2023-08-23

**Authors:** Bo Zhang, Pengtao Liu, Huakang Sheng, Yang Guo, Yongzhi Han, Lin Suo, Qipeng Yuan

**Affiliations:** State Key Laboratory of Chemical Resource Engineering, College of Life Science and Technology, Beijing University of Chemical Technology, Beijing 100029, China; 2020400268@buct.edu.cn (B.Z.); 2019400251@buct.edu.cn (P.L.); 2020400255@buct.edu.cn (H.S.); guoyang1016@163.com (Y.G.); 2022201299@buct.edu.cn (Y.H.); 2022500061@buct.edu.cn (L.S.)

**Keywords:** sulforaphene, oxidative damage, inflammation, Nrf2, anti−aging

## Abstract

Sulforaphene (SFE) is a kind of isothiocyanate isolated from radish seeds that can prevent free-radical-induced diseases. In this study, we investigated the protective effect of SFE on oxidative-stress-induced damage and its molecular mechanism in vitro and in vivo. The results of cell experiments show that SFE can alleviate D-gal-induced cytotoxicity, promote cell cycle transformation by inhibiting the production of reactive oxygen species (ROS) and cell apoptosis, and show a protective effect on cells with H2O2-induced oxidative damage. Furthermore, the results of mice experiments show that SFE can alleviate D-galactose-induced kidney damage by inhibiting ROS, malondialdehyde (MDA), and 4-hydroxyalkenals (4-HNE) production; protect the kidney against oxidative stress-induced damage by increasing antioxidant enzyme activity and upregulating the Nrf2 signaling pathway; and inhibit the activity of pro-inflammatory factors by downregulating the expression of Toll-like receptor 4 (TLR4)—mediated inflammatory response. In conclusion, this research shows that SFE has antioxidant effects, providing a new perspective for studying the anti−aging properties of natural compounds.

## 1. Introduction

Cruciferous plants contain a specific group of compounds called glucosinolates (GLs), which undergo myrosinase−catalyzed hydrolysis to produce isothiocyanates (ITCs) [1]. To date, more than 100 types of ITCs have been found. Based on the differences between their R−side chains, ITCs can be further divided into three categories: aliphatic, aromatic, and indole ITCs [2,3]. It has been reported that the main function of ITCs is to activate phase II detoxifying enzymes, increasing the level of glutathione and enhancing the antioxidant capacity [4,5]. Sulforaphene (SFE, Figure 1A), a naturally occurring ITC found in *cruciferous* vegetables such as radish seeds and broccoli, has attracted attention as a preventative compound [6]. It has been reported that SFE can enhance spatial learning and memory ability in rats and improve cognition ability by regulating the phosphoinositide 3−kinase (PI3K)/protein kinase B (Akt)/glycogen synthase kinase−3β (GSK−3β) pathway [7]. The growing interest in natural bioactive plant products as alternative and complementary medicines for the prophylactic and therapeutic treatment of a wide range of human diseases has provided new perspectives on the study of the anti−aging properties of natural compounds [8,9].

Population aging is a serious problem and a heavy burden in modern society, and the causes of aging are complex and mostly unknown. It has been reported that the balance between ROS production and antioxidant defense determines the degree of cellular oxidative stress and aging [10]. Excessive ROS production can cause a variety of adverse biological reactions, including lipid peroxidation, DNA damage, mitochondrial metabolism disorders, and apoptosis [11,12]. At the same time, the cellular antioxidant defense system can remove excess free radicals from the body [13]. Nuclear factor−erythroid 2−related factor 2 (Nrf2) plays an important role in regulating the redox balance of the body, regulating the cell defense against various pathological injuries of the kidney, and thus protecting the kidney from acute and chronic injuries [14,15,16]. Recent studies have shown that the Nrf2/antioxidant−responsive element (ARE)/heme oxygenase 1 (HO−1) pathway is a potential target for kidney protection in various drug−induced kidney injury models [17,18]. Therefore, the activation of the Nrf2 pathway is a potential mechanism for delaying the aging process.

Senescent cells are characterized by stable cell cycle arrest, morphological and metabolic changes, chromatin remodeling, altered gene expression, and the appearance of senescence−associated secretory phenotype (SASP) [19,20]. Senescent cells are known to produce a number of factors associated with inflammation, such as pro−inflammatory cytokines and chemokines [21]. It has been reported that bacterial lipopolysaccharide (LPS), which is one of the main components of the cell wall of Gram−negative bacteria, can activate toll−like receptor 4 (TLR4) on host cells to initiate a pro−inflammatory response, producing inflammatory factors such as tumor necrosis factor−α (TNF−α), interleukin−6 (IL−6), and interleukin−1 beta (IL−1β) [22,23]. The TLR4−myeloid differentiation factor 88 (MyD88)–NF−κB signal pathway has emerged as a hotspot in the study of inflammatory response, immunity, and cell−death−associated senescence injury [24]. Therefore, the TLR4–MyD88–NF−κB signal pathway may be a promising therapeutic target for the treatment of kidney damage and aging−related pathology during aging.

In this study, we researched the effect of SFE on the protection of oxidatively damaged cells via its effect on apoptosis, cell cycle, and mitochondrial function, and then by examining the expression of proteins associated with apoptosis and cell cycle. Hence, it is highly important to study natural antioxidants and anti−inflammatory substances to slow down aging.

## 2. Results

### 2.1. SFE Promoted Cell Growth and Inhibited ROS Generation

Hydrogen peroxide (H_2_O_2_) is widely used as an inducer in modeling oxidative stress [25]. In our study, H_2_O_2_ was used to establish a cellular model of oxidative stress. In past studies, senescence modeling has often used HFF and HacaT cells [26,27,28]. In our study, we also used the above cell lines for our senescence modeling in order to facilitate comparison with previous studies. Firstly, HFF and HaCaT cells were treated with different concentrations (0–500 μM) of H_2_O_2_ for 3 h. The cell viability was checked via CCK−8 assay. As shown in Figure 2A,C, the survival rate of HFF and HaCaT cells decreased significantly in a dose−dependent manner. In the 100 μM H_2_O_2_−treated group, cell viability decreased to 68.64 ± 1.55% and 66.15 ± 4.72%, respectively. As previous studies have reported, H_2_O_2_ damage cell models were established when cell viability was reduced to 50–70%. Thus, the treatment in which 100 μM H_2_O_2_ was administered for 3 h was chosen for further experiments. In recent years, several studies have suggested that metformin may have anti−aging potential [29,30]. Cells were treated with different concentrations of SFE or metformin hydrochloride (MET) for 24 h. As shown in Appendix A, SFE promoted HFF cell proliferation at concentrations of 1.25, 2.5, and 5 μmol/L and HaCaT cell proliferation at concentrations of 0.625, 1.25, and 2.5 μmol/L, while MET promoted HFF and HaCaT cell proliferation at a concentration of 2 mmol/L. Next, the potential protective effect of SFE against oxidative−stress−induced injury in HFF and HaCaT cells was evaluated. The cytotoxicity of H_2_O_2_ on HFF and HaCaT cells was observed after the cells were pretreated with different concentrations of SFE and 2 mmol/L MET for 24 h. As shown in Figure 2B,D, compared with the control group, cell viability was reduced in the H_2_O_2_ group. However, pretreatment with SFE attenuated H_2_O_2_−induced cell death in a dose−dependent manner. The protective effect of high doses of SFE (5 and 2.5 μmol/L) was similar to that of 2 mmol/L MET. Cell pretreatment with SFE or MET increased cell activity. In order to evaluate the effect of SFE on H_2_O_2_−induced oxidative stress in HFF and HaCaT cells, ROS levels were measured according to DCFH−DA fluorescence using confocal microscopy. The results show that the fluorescence intensity of the H_2_O_2_−treated group was significantly higher than that of the control group. Then, with the increase in SFE concentration, the intracellular ROS activity gradually decreased. Treatment with SFE or MET alone did not cause an increase in intracellular ROS (Figure 2E). The morphological effects of apoptosis were observed via DAPI staining. The cells treated with H_2_O_2_ showed chromatin concentration and nuclear contraction. However, pre−treatment with SFE can alleviate the above symptoms (Figure 2F). These results suggest that pre−treatment with SFE can inhibit H_2_O_2_−induced ROS production and protect cells from intracellular ROS damage. 

### 2.2. SFE Alleviated H_2_O_2_−Induced Mitochondrial Damage in HFF and HaCaT Cells

Mitochondria are important for energy and ROS production and maintaining the normal physiological function of cells [31]. Excessive ROS leads to the destruction of mitochondrial function, resulting in oxidative damage [32]. To evaluate the mitochondrial function of HFF and HaCaT cells, ΔΨm and ATP were first measured after SFE and H_2_O_2_ treatment. As shown in Figure 3A, the green fluorescence intensity of the JC−1 monomer in HFF and HaCaT cells after H_2_O_2_ treatment was remarkably higher than that of the control group. The conversion of JC−1 from red to green fluorescence has been used as an indicator of early apoptosis. However, SFE significantly increased the production of JC−1 aggregates in a concentration−dependent manner. The intensity of red mitochondrial fluorescence after MET and H_2_O_2_ treatment was less pronounced than in the SFE−treated group. SFE or MET alone did not result in a decrease in intracellular ΔΨm. The ATP expression levels in HFF and HaCaT cells were further examined. It was shown that H_2_O_2_ treatment resulted in a decrease in intracellular ATP levels, whereas the SFE pretreatment of HFF and HaCaT cells gradually increased ATP expression (Figure 3B,C). In conclusion, these results suggest that the protective effect of SFE is related to reductions in mitochondrial dysfunction and mitochondrial−damage−induced apoptosis.

### 2.3. SFE Attenuated Apoptosis in H_2_O_2_−Treated HFF and HaCaT Cells

To detect the anti−apoptosis effect of SFE, flow cytometry was used to detect the apoptosis of cells treated with H_2_O_2_. As shown in Figure 4A–D, the percentage of apoptotic cells in the H_2_O_2_−treated group was significantly increased compared with the control group. However, pretreatment with SFE reduced the percentage of apoptotic cells in a dose−dependent manner. The addition of SFE or MET alone did not cause excessive apoptosis, and pretreatment with MET reduced the percentage of apoptotic cells. The apoptosis rate was the lowest when SFE concentrations were 5 and 2.5 μmol/L. Furthermore, the expression level of apoptotic−related proteins in HFF and HaCaT cells was examined via western blotting analysis. Compared with the control group, there were significant increases in cleaved caspase−9/caspase−9 and cleaved caspase−3/caspase−3 in the H_2_O_2_ group. However, SFE pretreatment downregulated the expression levels of cleaved caspase−9 and cleaved caspase−3 (Figure 5A–D). The results show that SFE has a strong cytoprotective effect on HFF and HaCaT cells with H_2_O_2_−induced damage and can protect cells from damage by inhibiting the apoptosis signal pathway. 

### 2.4. Effect of SFE on the Cell Cycle in HFF and HaCaT Cells with H_2_O_2_−Induced Damage

Cell growth inhibition is usually associated with cell cycle arrest at a specific stage. In order to investigate the protective mechanism of SFE against oxidative damage, the effect of H_2_O_2_−induced oxidative damage on the cell cycle was detected via flow cytometry. As shown in Figure 6A,B, compared with the control group, the percentage of HFF cells with H_2_O_2_−induced damage that were in the G_0_/G_1_ phase increased from 52.83 ± 4.73% to 75.02 ± 2.28%, and those in the S−phase decreased from 35.88 ± 3.26% to 12.75 ± 1.26%, suggesting that the HFF cells in the H_2_O_2_ group were arrested in G_0_/G_1_. However, the SFE pretreatment group showed a dose−dependent decline, with no difference in cell cycle between the highest−dose group and the control group. MET pretreatment inhibited the cell cycle and promoted cell growth in the injured group. The results showed that H_2_O_2_−induced DNA damage kept HFF cells in the G_0_/G_1_ phase and restricted their entry into the S phase for DNA replication, thus inhibiting cell proliferation. To further investigate the effect of SFE on the cell cycle, western blotting was used to detect the expression of Cyclin D1 and CDK6. As shown in Figure 7A,B, the protein expression levels of Cyclin D1 and CDK6 in the H_2_O_2_ group were significantly reduced compared to those in the control group. However, pretreatment with SFE gradually increased the levels of these proteins in H_2_O_2_−treated HFF cells. MET had no significant effect on the protein expression levels of Cyclin D1 and CDK6. In addition, the percentage of HaCaT cells with H_2_O_2_−induced damage in the G_0_/G_1_ phase ranged from 80.20 ± 1.99% to 41.79 ± 4.51%, and the percentage in the S phase ranged from 11.16 ± 1.45% to 47.9 ± 2.43%, indicating that cells were blocked in the S phase (Figure 6C,D). However, the S phase block was reduced after SFE pretreatment, indicating that SFE can protect HaCaT cells from oxidative damage and promote cell proliferation. As shown in Figure 7C,D, the expression levels of Cyclin A and CDK2 in the SFE−pretreated group gradually increased compared with those in the H_2_O_2_−treated group. MET pretreatment significantly increased the expression of the cycle−related protein, and SFE or MET treatment alone showed no significant difference from the normal group. These results suggest that SFE alleviates the proliferation of damaged cells by accelerating cell cycle processes.

### 2.5. Effect of SFE on D−Galactose (D−gal)−Induced Kidney Injury in Mice

As an animal ages, the weight of a damaged organ changes, and so does the organ index. For example, the brain, spleen, kidney, and liver are important indicators of human aging [33]. Aging usually causes weight loss in most organs and affects the body’s immune and metabolic abilities [34]. To evaluate the potential protective efficacy of SFE in mice with D−gal−induced kidney damage, the mice were fed with SFE (10, 30, and 50 mg/kg b.w.) by gavage daily for 8 weeks. First, we recorded the initial and final weights of each group of mice. As shown in Table 1, the mice in the D−gal group gained less weight than those in the control group. However, the weight gain of SFE−treated mice was dose−dependent, and the weight gain of β−nicotinamide mononucleotide (NMN) −treated mice was equivalent to that of the SFE−treated mice (30 mg/kg b.w.). At the same time, the kidney index is also shown in Table 1. The kidney index of the D−gal treatment group was significantly lower than that of the control group, the kidney index of the SFE treatment group gradually rose to be higher than that of the D−gal treatment group, and the kidney index of the NMN group was slightly higher than that of the D−gal treatment group, with no significant difference.

### 2.6. SFE Ameliorated D−gal−Induced Oxidative Kidney Damage in Mice

H&E stains of the kidneys are shown in Figure 8A. Compared with the control group, after D−gal injections for 8 weeks, the number of functional glomeruli in the kidneys of mice was significantly reduced, and the lumens of glomerulus were enlarged. Renal cells also experienced edema, degeneration, vacuolation, inflammatory cell infiltration, and fibrosis. After treatment with SFE and NMN, the kidney structure damage was improved significantly, the kidney tubules were arranged tightly, and the structure was obviously restored. These results suggest that SFE can alleviate the kidney tissue structure damage caused by D−gal and play a protective role in the kidney. The levels of ROS and the lipid peroxidation markers 4−HNE and malondialdehyde (MDA) were increased in the kidney cells of the D−gal−induced mouse aging model. In this study, the results showed that D−gal treatment significantly elevated the levels of ROS and oxidation products including 4−HNE and MDA. However, treatment with SFE and NMN prevented the elevation of these biomarkers in the kidneys of D−gal−treated mice (Figure 8B–D). To further explore the antioxidative properties of SFE, the activities of total antioxidant capacity (T−AOC), CAT, and SOD were investigated. As shown in Figure 8E–G, D−gal treatment significantly decreased the activities of T−AOC, CAT, and SOD compared with the control group, whereas the declines of these enzyme activities were gradually reversed with an increased dose of SFE. These results suggest that SFE alleviates oxidative stress by reducing lipid oxidation levels and increasing antioxidant enzyme activity.

### 2.7. SFE Enhanced the Antioxidant Capacity of the Kidney by Activating the Nrt2–ARE Signal Pathway

The Kelch−like ECH−associated protein l (Keap1)/Nrf2/ARE signal pathway is the main pathway involved in cellular defense and resistance to endogenous and exogenous stress and activates a variety of antioxidant genes and phase II detoxifying enzymes to maintain redox balance [35,36]. The expressions of Nrf2 and its downstream proteins were examined via western blot. As shown in Figure 9A,B, the expression level of Nrf2 in the SFE group gradually increased compared with that in the D−gal−treated group. In addition, the ability of NMN to increase the Nrf2 protein level was higher than that of SFE. Nrf2 enters the nucleus after activation and binds to ARE with the assistance of Maf proteins. It activates downstream antioxidant proteins including HO−1, quinone oxidoreductase−1 (NQO−1), and glutamate−cysteine ligase modifier (GCLM); scavenges free radicals; and protects mice from oxidative damage [37,38]. Subsequently, the expression levels of downstream proteins HO−1, NQO1, and GCLM were detected. As shown in Figure 9C–E, the expression levels of HO−1, NQO1, and GCLM in the D−gal treatment group were decreased compared with the control group. However, the expression levels of these proteins were significantly increased after SFE treatment compared to after D−gal treatment, and the expression levels of these proteins were higher in the NMN treatment group than in the SFE treatment group. In summary, SFE upregulates the expression of detoxification enzymes and antioxidant enzymes through the Nrf2–ARE signal pathway, inhibits the production of ROS and lipid oxides, and thus protects the kidney from oxidative damage.

### 2.8. SFE Attenuated D−gal−Induced Inflammatory Responses in the Kidneys of Mice

TLR4 is a major component of the LPS−recognition receptor complex that induces the production of pro−inflammatory cytokines such as TNF−α, IL−6, and IL−1β [39]. To determine the inhibitory effect of SFE on D−gal−induced inflammation in aging mice, we measured the expression of inflammatory factors in kidney tissue. These results showed that the D−gal treatment significantly increased the LPS level, while the SFE treatment significantly inhibited the increase in LPS in D−gal treated mice (Figure 10A). In addition, the levels of IL−1β, IL−6, interferon−gamma (IFN−γ), TNF−α, CXC chemokine ligand−10 (CXCL10), and monocyte chemotactic protein−1 (MCP−1) in D−gal−treated mice were significantly increased, while these levels were significantly decreased in SFE−treated mice, and the inflammation level was decreased after NMN treatment, though this was not as obvious as it was in SFE−treated mice (Figure 10B–G). Compared with the control group, the expression levels of interleukin−10 (IL−10) and transforming growth factor−β (TGF−β) decreased after D−gal treatment and increased in a dose−dependent manner after SFE treatment, and the anti−inflammatory ability of NMN after D−gal treatment was higher than that after SFE treatment (Figure 10H,I). The protein expression levels of mice kidney−tissue−related proteins were measured via western blotting, including TLR4, MyD88, NF−κB p−p65, p−IκB, and IL−6. D−gal treatment markedly upregulated the expression levels of TLR4, MyD88, NF−κB p−p65, p−IκB, and IL−6 compared with the control group. However, SFE treatment at a dose of 50 mg per kg b.w. significantly alleviated the increase in TLR4, MyD88, phospho−NF−κB p65 (p−NF−κB p65), phospho−IκB (p−IκB), and IL−6 expression in the kidneys of D−gal−treated mice (Figure 11A–F). Taken together, these results suggest that SFE inhibits D−gal−induced inflammatory responses by inhibiting the TLR4/MyD88 pathway, thereby alleviating kidney damage in aging mice.

## 3. Discussion

Oxidative stress is the excessive accumulation of ROS, resulting in an imbalance between oxidants and antioxidants [40]. Excessive ROS can lead to damage to DNA, enzymes, and membranes, contributing to the pathogenesis of kidney injury [41,42]. Epidemiological studies have shown that broccoli, radish, and other *cruciferous* vegetables have anti-inflammatory, anticancer, and antioxidant effects, as well as other functions [43,44]. The prominent anticancer effects of *cruciferous* vegetables are mainly attributed to their abundant content of isothiocyanate compounds, such as SFE and sulforaphane (SFN, Figure 1B) [45]. SFE is the main isothiocyanate produced by the myrosinase hydrolysis of glucoraphanin in *cruciferous* vegetables such as radish seeds [46]. SFN is derived mainly from broccoli and its seeds [47]. Studies have shown that SFN has anti−inflammation, anti−bacteria, and anti−aging effects and can improve the body’s antioxidant capacity and immunity [48,49,50]. Some studies have reported that SFE was found have several important biological effects, such as anti−inflammatory and anticancer effects [41,51]. In this study, we explored the anti−aging effect of SFE in vitro and in vivo. Furthermore, the potential molecular mechanism of SFE anti−aging was studied. The results showed that SFE blocked oxidative stress and mitochondrial dysfunction induced by excess ROS and reduced the apoptotic cascade. In vivo, SFE reduces oxidative stress by increasing antioxidant enzyme activity and upregulating the Nrf2 signal pathway. Our results suggest that SFE has therapeutic potential as a targeted agent for the treatment of aging.

H_2_O_2_ is the main precursor of highly reactive oxygen radicals such as superoxide anion and hydroxyl radical, which can induce oxidative stress damage and cell apoptosis [52]. In this study, we found that SFE pretreatment increased cell viability and mitigated cell morphological changes in H_2_O_2_−treated HFF and HaCaT cells (Figure 2A–D,F). It is well known that the overproduction of ROS in mitochondria leads to mitochondrial damage, which in turn leads to the release of pro−apoptotic substances in mitochondria and ultimately induces apoptosis [53,54]. Our study found that SFE inhibited ROS production and reduced the percentage of apoptotic cells (Figure 2E and Figure 4A–D). Meanwhile, western blot analysis showed that SFE reduced the expression of cleaved caspase−9/caspase−9 and cleaved caspase−3/caspase−3, and alleviated apoptosis (Figure 5A–D). It has been reported that the intracellular ATP level plays a crucial role in cell survival, and a reduction in ATP level can also lead to cell apoptosis [55]. In this study, we found that pretreatment with SFE can increase ATP levels and decrease apoptotic protein expression in H_2_O_2_−treated cells (Figure 3B,C and Figure 6A,C). In summary, our results suggest that SFE can alleviate ROS−mediated apoptosis in HFF and HaCaT cells. 

Mitochondrial morphology is closely related to mitochondrial function and metabolic activities [56]. ΔΨm is closely related to the production of ATP, which is a prerequisite for maintaining the oxidative phosphorylation of mitochondria and producing ATP. When oxidative damage occurs in cells, it leads to mitochondrial dysfunction, resulting in insufficient ATP production [57,58]. In this study, we found that pretreatment with SFE can increase ΔΨm and mitigate ATP reduction in H_2_O_2_−treated HFF and HaCaT cells (Figure 3A–C). ATP and ROS are produced during mitochondrial oxidative phosphorylation to maintain homeostasis, and changes in ATP and ROS levels can affect cell proliferation [59]. ROS can regulate the cell cycle, and hyper oxygen can block cell proliferation and inhibit cell proliferation during the G_1_, S, and G_2_ phases [60]. Besides apoptosis, compared with the control, H_2_O_2_ treatment resulted in HFF cell cycle arrest at G_0_/G_1_ (Figure 6A,B). However, the cycles of HaCaT cells treated with H_2_O_2_ were blocked in the S phase (Figure 6C,D). Pretreatment with SFE reduced cell cycle arrest in a concentration−dependent manner. Cyclin is an important part of the cell cycle mechanism. Type−D and −E cyclins are required for the normal progression of the G_1_ phase of the cell cycle, with type−D cyclins primarily activating Cdk4 and −6 and cyclin E activating Cdk2 [61,62]. In the present study, the protein expression levels of Cyclin D1 and CDK6 were significantly reduced in the H_2_O_2_ group, whereas these levels were reversed by SFE pretreatment (Figure 7A,B). Meanwhile, the protein expression levels of Cyclin A and CDK2 were increased by SFE pretreatment (Figure 7C,D). Taken together, these results suggest that H_2_O_2_ can cause cell cycle arrest, and SFE can protect cells from oxidative damage and alleviate cell cycle arrest by inhibiting ROS production.

It has been reported that mice experiencing D−gal−induced aging have similar symptoms to naturally aging mice, so D−gal is widely used in anti−aging pharmacology [63]. We established a D−gal−induced aging mouse model, and after the experiment, both body weight and organ coefficient were reduced (Table 1), which was consistent with the reported results [64]. Excess galactose is metabolized to galactoalcohol and hydrogen peroxide in the body, which leads to ROS accumulation, causing oxidative stress and inflammatory reactions [65]. In this study, we found that SFE treatment can improve D−gal−induced kidney injury in mice, as histopathological examination showed (Figure 8A). Excess ROS causes lipid peroxidation, protein denaturation, DNA damage, and oxidative damage to other biomolecules, which result in harmful effects on cells and tissues [66]. The lipid peroxidation products produced in the metabolism of oxygen free radicals in MDA and 4−HNE can reflect the existence of lipid peroxidation free radicals in the system and the degree of reaction [67,68]. The results demonstrate that treatment with SFE inhibits D−gal−induced ROS generation and the production of MDA and 4−HNE (Figure 8B–D). However, excess ROS can be eliminated by kidney cell antioxidant defenses induced by both enzymatic and non−enzymatic mechanisms [69,70]. In the antioxidant defense system, the enzymes CAT, GSH−Px, and SOD can clear oxygen−free radicals [71]. In the present study, SFE effectively reversed the downregulation of T−AOC, SOD, and CAT activities in D−gal−treated mice (Figure 8E–G), indicating the improvement of the antioxidant defense system. The Nrf2 antioxidant pathway is one of the most important antioxidant systems in the body, which enhances the endogenous antioxidant capacity by regulating the expression of antioxidative protease and phase II detoxifying enzyme genes [72,73]. Our results show that SFE can restore Nrf2 activity in the kidneys of D−gal−treated mice and subsequently upregulate the expression levels of downstream enzymes such as HO−1, NQO1, and GCLM (Figure 9A–E). These results suggest that SFE protects the kidney from oxidative damage by activating Nrf2 and the resulting antioxidant defense system. Some studies have shown that SFN works against oxidative stress primarily through Nrf2 activation and the subsequent production of antioxidant proteins and phase II detoxifying enzymes [74,75,76]. Moreover, SFN exerted renoprotective effects against various kidney injury models. Khaleel et al. [77] found that SFN improved streptozotocin−induced diabetic nephropathy by upregulating Nrf2 and HO−1 genes and downregulating IL−6 and caspase−3 genes. In general, the protective effects of SFE and SFN on the kidney is mainly through the Nrf2 signaling pathway.

Recent studies have shown that SFE has antioxidant and anti−inflammatory properties and can reduce oxidative stress and inflammatory responses [78,79]. TLR4 is a major component of the LPS recognition receptor complex. Activation of the TLR4/MyD88 signal pathway triggers intracellular signaling cascades that ultimately lead to NF−κB activation, which induces pro−inflammatory cytokines [80]. TNF−α, IL−1β, and IL−6 are considered pro−inflammatory cytokines that cause inflammation in various tissues and organs and can accelerate cell aging [81]. Our results show that SFE can ameliorate the inflammatory cytokines LPS, TNF−α, IL−6, and IL−1β (Figure 10A–C,E). Cell senescence is often accompanied by the generation of SASP. SASP is composed of proinflammatory cytokines, growth factors, chemokines, and matrix remodeling enzymes, which can cause chronic low−grade inflammation and disease in the body [82,83]. The levels of chemokines (CXCL10 and MCP−1) and IFN−γ in SFE−treated D−gal mice were reduced in a concentration−dependent manner (Figure 10D,F,G). IL−10 is known as a cytokine with anti−inflammatory and immunosuppressive properties [84]. In the present study, SFE effectively reversed the downregulation of IL−10 and TGF−β levels in D−gal−treated mice (Figure 10H,I). Furthermore, the protein expression levels of TLR4, MyD88, NF−κB p−p65, p−IκB, and IL−6 were significantly increased in the D−gal group, whereas these levels were reversed by SFE treatment (Figure 11A–F). Taken together, these results suggest that SFE attenuates D−gal−induced renal injury by suppressing the inflammatory response in mice.

## 4. Materials and Methods

### 4.1. Materials

H_2_O_2_ (35 wt%) was purchased from Sigma−Aldrich (St. Louis, MO, USA). D−gal (≥99.0%) was obtained from Solarbio (Beijing, China). NMN (≥98.0%) and MET (≥98.0%) were obtained from Yuanye Biotechnology Co., Ltd. (Shanghai, China). According to a previously used method, SFE was isolated and purified [85,86]. Primary antibodies against cleaved caspase−9, caspase−9, cleaved caspase−3, caspase−3, Cyclin D1, CDK6, Cyclin A, CDK2, Nrf2, HO−1, NQO−1, GCLM, TLR4, MyD88, p−NF−κB p65, p−IκB, and IL−6 were purchased from Cell Signaling Technology (Danvers, MA, USA). An antibody against β−actin was purchased from Santa Cruz Biotechnology (Santa Cruz, CA, USA). Secondary antibody HRP−conjugated goat anti−rabbit IgG and goat anti−mouse IgG were obtained from Abcam (Cambridge, MA, USA). Maintenance feed for mice (#SPF−F02−001) was purchased from SPF (Beijing) Biotechnology Co., Ltd.

### 4.2. Cell Culture and Treatments

HFF (#FH0794, human fibroblast line) and HaCaT (#FH0186, immortalized aneuploid human keratinocyte cell line) were purchased from Shanghai Fuheng Biotechnology Co., Ltd. (Shanghai, China) and were cultured at 37 °C and 5% CO_2_ in Dulbecco’s Modified Eagle medium (Hyclone, Logan, Utah, USA). SFE stock solution was prepared with DMSO at 50 mmol/L, divided into 1.5 mL centrifuge tubes, and stored at −80 °C. The specific experimental steps were as follows: HFF and HaCaT cells were inoculated in 96−well plates and treated with different concentrations of SFE or MET for 24 h, and cell activity was detected using a cell counting kit−8 (CCK−8, Dojin Laboratories, Kumamoto, Japan). Cell proliferation was detected using a slightly modified version of the CCK8 method [87]. Cells were treated with different concentrations of H_2_O_2_ for 3 h, and CCK−8 was added to detect cell activity.

### 4.3. Detection of Intracellular ROS Production

Total intracellular ROS levels were detected by fluorescent probe DCFH−DA [88]. Cells were cultured in 6−well plates for 24 h and treated with different concentrations of SFE or MET for 24 h, then treated with H_2_O_2_ for 3 h. A 10 μM fluorescent probe and 2′,7′−dichlorofluorescein−diacetate (DCFH−DA, Beyotime, Shanghai, China) were added to the cell, which were then incubated in the dark at 37 °C for 30 min. After incubation, the cells were washed with PBS and immediately observed under a fluorescence microscope (Nikon Corp., Tokyo, Japan) within 1 h.

### 4.4. Analysis of Nuclear Morphology

DAPI is a fluorescent dye that binds strongly to DNA and is used to detect the morphology of the cell nucleus, which is often used in fluorescence microscopy [89]. Cells were cultured in 6−well plates for 24 h, different concentrations of ligands were added, 1mL DAPI was added according to the instructions, and the cells were placed at room temperature for 5 min and washed with PBS. The cell nuclei were detected via fluorescence microscopy (Nikon Corp., Tokyo, Japan).

### 4.5. Determination of Mitochondrial Membrane Potential (Δψm)

The changes in the mitochondrial membrane potential after treatment were monitored using a JC−1 probe. When the mitochondrial membrane potential is high, the polymerization produces a red fluorescence, and when the membrane potential is low, the JC−1 produces a green fluorescence as a monomer [90]. The JC−1 fluorescent probe (Beyotime, Shanghai, China) was used to detect mitochondrial membrane potential. Cells were cultured in 6−well plates and treated with different ligands. According to the instructions, the JC−1 dye solution was incubated for 30 min, washed with PBS, and observed under a fluorescence microscope (Nikon Corp., Tokyo, Japan).

### 4.6. Intracellular ATP Level Detection

A modified version of the method given by Li et al. for detecting intracellular ATP content was used [91]. Cells were cultured in 6−well plates, different concentrations of ligands were added, and the intracellular ATP level was detected according to the instructions of the ATP detection kit (Beyotime, Shanghai, China).

### 4.7. Measurement of Apoptotic Cells

Cells in different stages of apoptosis were analyzed via modified flow cytometry [92]. The cells were cultured in 6−well plates, and the experiments were performed according to the previous steps. Annexin V−fluorescein isothiocyanate (Annexin V−FITC)/propidium iodide (PI) double staining (Beyotime, Shanghai, China) was used to detect the apoptosis of HFF and HaCaT cells, and flow cytometry (BD Pharmingen, San Diego, CA, USA) was used to detect the cells in time.

### 4.8. Cell Cycle Analysis

Cell cycle analysis was performed via flow cytometry, with minor modifications according to the method of Xu et al. [93]. The cells were treated with different concentrations of SFE or MET in a 6 cm petri dish for 24 h and incubated with H_2_O_2_ for 3 h. After incubation, the cells were digested with trypsin, then frozen and fixed with 70% ethanol for 24 h, then washed with pre−cooled PBS 3 times, and incubated with RNaseA at 37 °C for 30 min. Finally, PI was added, and the cells were incubated at room temperature for 30 min. The samples were detected using flow cytometry. Data were analyzed by ModFit 5.0 software (BD Biosciences, San Jose, CA, USA). The percentage of cells in the G_0_/G_1_ phase, S phase, and G_2_/M phase were analyzed.

### 4.9. Animal Experiment

Six−week−old male C57BL/6J mice were purchased from Beijing Vital River Laboratory Animal Technology Co., Ltd. (Beijing, China). The protocols used for all animal studies were approved by the Beijing Viewsolid Biotechnology Co., Ltd. (Beijing, China). Animal Policy and Welfare Committee and complied with the NIH guidelines (guide for the care and use of laboratory animals). Throughout the experiment, all the mice were fed with enough food and water. After 1 week of adaptation, the mice were randomly divided into 6 groups. Group 1 was the control group. Group 2 was the D−gal group, in which from Day 1 to Day 14, mice were injected subcutaneously with D−gal at a dose of 300 mg/kg b.w.; from Day 15 to Day 42, the D−gal dose was 500 mg/kg b.w.; and from Day 43 to Day 56, the D−gal dose was 800 mg/kg b.w. Group 3 was the D−gal + SFE group(10 mg/kg b.w.), Group 4 was the D−gal + SFE group (30 mg/kg b.w.), Group 5 was the D−gal + SFE group (50 mg/kg b.w.), and Group 6 was the D−gal + NMN group (positive drug, 100 mg/kg b.w.). In addition, D−gal was administered subcutaneously to the control group 2 h after administration. In this experiment, three different concentrations of SFE were set up to observe the protective effect of SFE on D−gal−induced aging in mice. NMN was selected as the positive control. The experiment lasted for 8 weeks. On the 57th day, all the mice were euthanized, and fresh blood and kidney tissue were collected and stored at −80 °C for future testing.

### 4.10. Histopathological Examination of the Kidney

H&E staining is a widely used, routine staining technique in histopathology, which can observe the overall morphology of tissues [94]. Kidney sections were fixed with 4% paraformaldehyde. After 24 h, the kidney tissues were embedded with paraffin wax. The samples were then cut into 5 µm slices and stained with hematoxylin−eosin (H&E). An optical microscope (Nikon, Tokyo, Japan) observed the sections.

### 4.11. Biochemical Analysis

The levels of ROS, 4−HNE, MDA, T−AOC, CAT, SOD, LPS, IL−1β, IL−6, IFN−γ, TNF−α, CXCL10, MCP−1, IL−10, and TGF−β in the kidney tissues were detected using ELISA kits (eBioscience, San Diego, CA, USA). A BCA protein assay kit (Nanjing Jiancheng Bioengineering Institute, Nanjing, China) was used for total protein determination, described as pg protein per mL.

### 4.12. Western Blotting Analysis

The target proteins in cells and tissues were quantified using the modified WB method [95]. Cells and kidney tissues were harvested and lysed for 20 min in RIPA buffer. Protein concentration was detected using BCA reagent, and the proteins (30 µg) were separated using 12% SDS−PAGE. After electrophoresis, protein samples were transferred to PVDF membranes (Millipore, Billerica, MA, USA). Then, the membranes were blocked with TBST containing 5% milk for 1 h. The membranes were irradiated at 4 °C with primary antibodies overnight. The next day, the membranes were cleaned with TBST and incubated with a secondary antibody for 1.5 h. After TBST washing, observations were made using enhanced chemiluminescence ECL reagent (Thermo Scientific, MA, USA), and a determination was made using an imaging system (Shanghai Tianneng Technology Co., Ltd.).

### 4.13. Statistical Analysis

All measurements were triplicate. All data were expressed as mean ± standard deviation (S.D.). Statistical analysis was performed using Prism GraphPad 8.0.2 (GraphPad Software Inc., San Diego, CA, USA). Significant differences among all groups were calculated according to one−way ANOVA and Bonferroni post hoc test. *p*−value < 0.05 was considered statistically significant.

## 5. Conclusions

Aging is now a pressing issue in global health; however, drug therapy comes with frequent toxic side effects [96]. In conclusion, this study reveals that SFE can mitigate cytotoxicity and demonstrate protective effects by improving mitochondrial function, reducing apoptosis, and accelerating cell cycle progression. SFE downregulates ROS production, thereby inhibiting MDA and 4−HNE production in the kidney. In addition, SFE can protect against oxidative−stress−induced kidney injury by upregulating the expression of Nrf2 protein and its downstream antioxidant proteins and phase II detoxifying enzymes. Additionally, SFE can inhibit the level of SASP factors in the kidney by downregulating the TLR4–MyD88 signal pathway. Therefore, SFE has effective kidney protective activity and may provide a promising approach to the prevention of kidney injury.

## Figures and Tables

**Figure 1 ijms-24-13129-f001:**
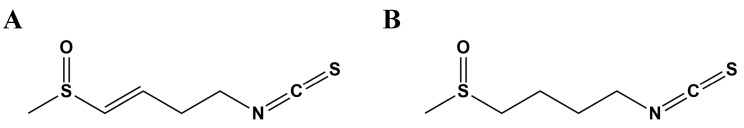
(**A**) The chemical structure of sulforaphene. (**B**) The chemical structure of sulforaphane.

**Figure 2 ijms-24-13129-f002:**
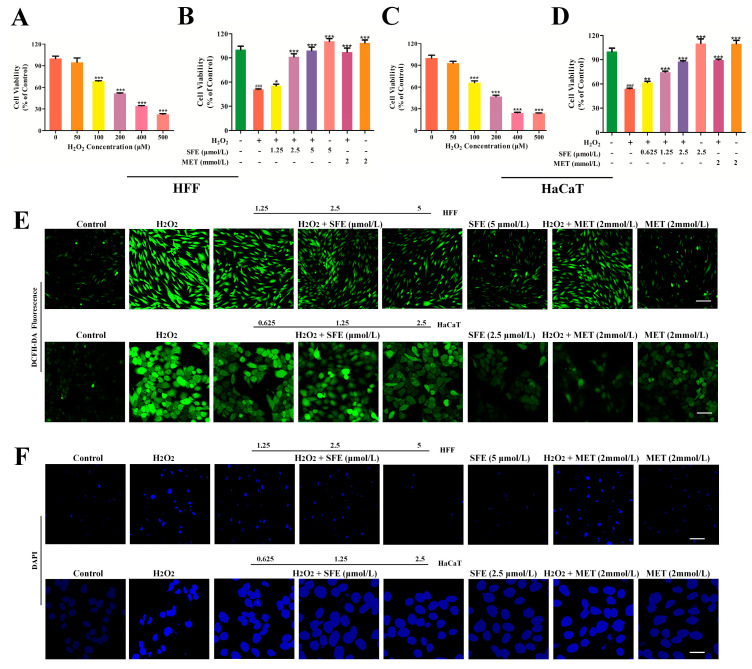
SFE promoted cell growth and inhibited ROS generation. (**A**) HFF cells were treated with different concentrations of H_2_O_2_ for 3 h (*** *p* < 0.001). (**B**) Effect of SFE on cell viability in HFF cells with H_2_O_2_-induced damage at the concentrations of 1.25, 2.5, and 5 μmol/L. (**C**) HaCaT cells were treated with different concentrations of H_2_O_2_ for 3 h (*** *p* < 0.001). (**D**) Effect of SFE on cell viability in HaCaT cells with H_2_O_2_-induced damage at the concentrations of 0.625, 1.25, and 2.5 μmol/L. (**E**) DCFH-DA was used to stain the cells, and the fluorescence intensity of DCF was detected using a fluorescence microscope (Bar = 50 μm). (**F**) In HFF and HaCaT cells treated with SFE and MET for 24 h and 100 μM H_2_O_2_ for 3 h, morphological changes were observed under a fluorescence microscope (Bar = 50 μm). Data are shown as mean ± S.D. (*n* = 5) for each group. ^###^
*p* < 0.001 vs. control group; * *p* < 0.05, ** *p* < 0.01, *** *p* < 0.001 vs. H_2_O_2_−treated group.

**Figure 3 ijms-24-13129-f003:**
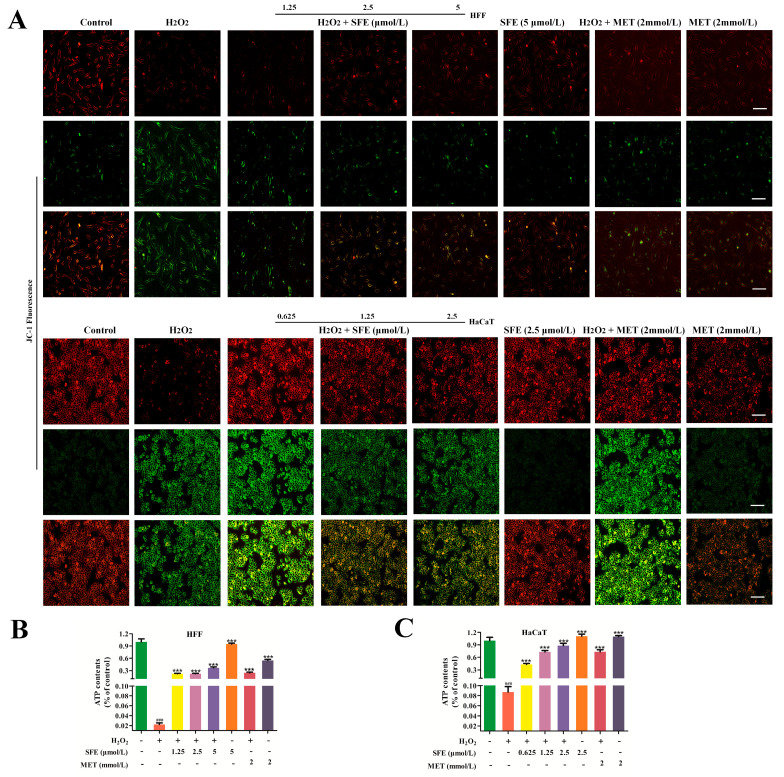
SFE alleviated H_2_O_2_-induced mitochondrial damage in HFF and HaCaT cells. (**A**) The ΔΨm was detected using a JC−1 mitochondrial membrane potential assay kit (Bar = 50 μm). JC−1 aggregates produced red fluorescence. JC−1 monomers produced green fluorescence. (**B**) The ATP levels in the HFF cells were determined by pretreatment with different amounts of SFE, 2 mmol/L MET for 24 h, or H_2_O_2_ for 3 h. (**C**) The ATP levels in the HaCaT cells were determined by pretreatment with different amounts of SFE, 2 mmol/L MET for 24 h, or H_2_O_2_ for 3 h. Data are shown as mean ± S.D. (*n* = 5) for each group. ^###^
*p* < 0.001 vs. control group, *** *p* < 0.001 vs. H_2_O_2_-treated group.

**Figure 4 ijms-24-13129-f004:**
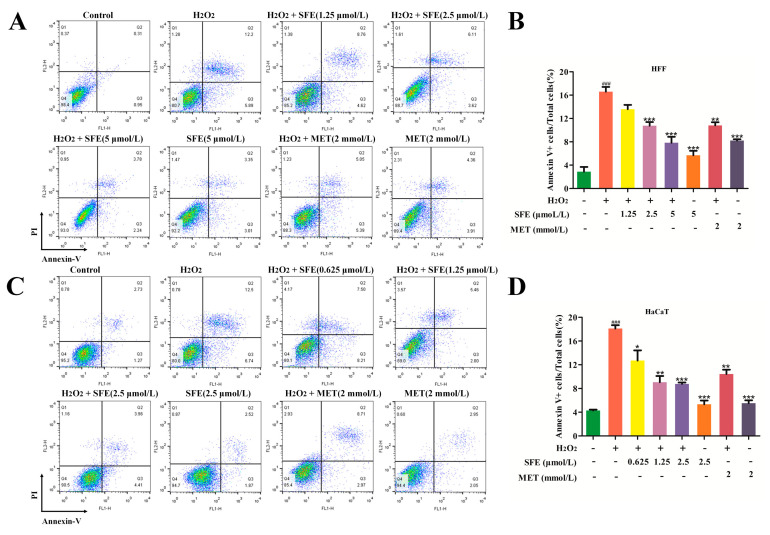
SFE attenuated apoptosis in H_2_O_2_-treated HFF and HaCaT cells. (**A**) Cell apoptosis in HFF cells was detected via flow cytometry. (**B**) The apoptosis rates of HFF cells. (**C**) Cell apoptosis in HaCaT cells was detected via flow cytometry. (**D**) The apoptosis rates of HaCaT cells. Data are shown as mean ± S.D. (*n* = 5) for each group. ^###^
*p* < 0.001 vs. control group, * *p* < 0.05, ** *p* < 0.01, *** *p* < 0.001 vs. H_2_O_2_−treated group.

**Figure 5 ijms-24-13129-f005:**
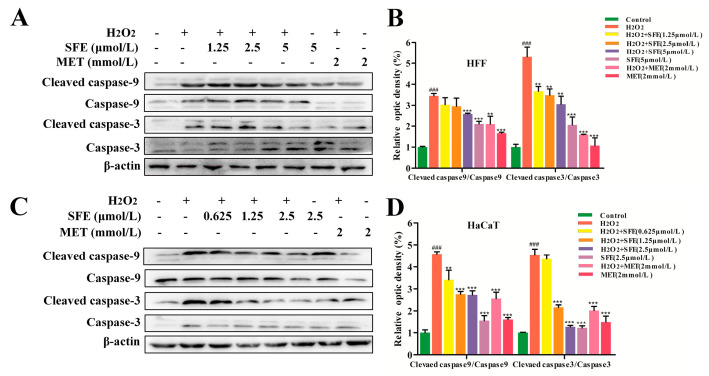
SFE inhibited apoptotic−related proteins in H_2_O_2_-treated HFF and HaCaT cells. (**A**) The expression levels of cleaved caspase-9, caspase-9, cleaved caspase-3, and caspase-3 in HFF cells were detected via western blotting analysis. (**B**) Quantification of cleaved caspase-9/caspase-9 and cleaved caspase-3/caspase-3 expression. (**C**) The expression levels of cleaved caspase-9, caspase-9, cleaved caspase-3, and caspase-3 in HaCaT cells were detected via western blotting analysis. (**D**) Quantification of cleaved caspase-9/caspase-9 and cleaved caspase-3/caspase-3 expression. Data are shown as mean ± S.D. (*n* = 5) for each group. ^###^
*p* < 0.001 vs. control group, ** *p* < 0.01, *** *p* < 0.001 vs. H_2_O_2_-treated group.

**Figure 6 ijms-24-13129-f006:**
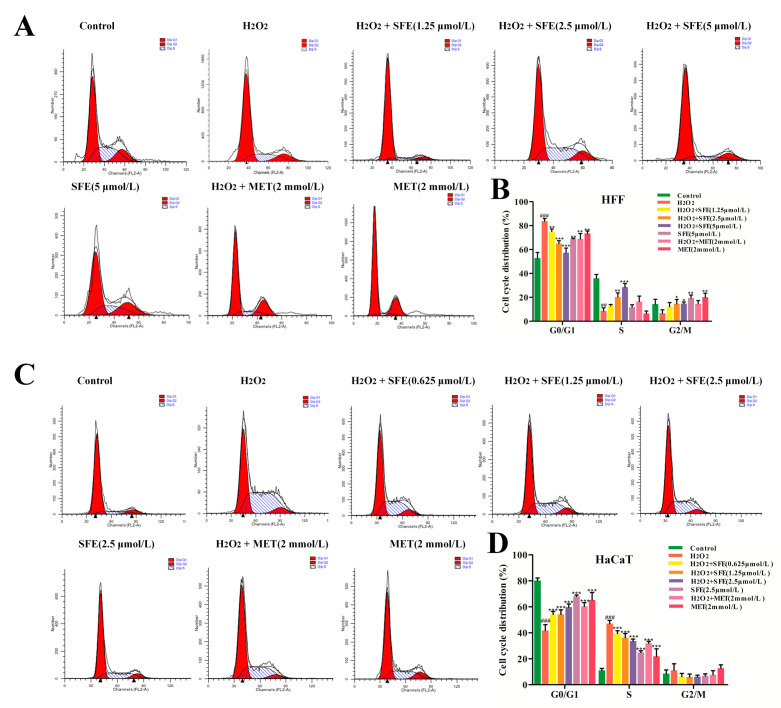
Effect of SFE on the cell cycle in HFF and HaCaT cells with H_2_O_2_−induced damage. (**A**) Cycle distribution of HFF cells was determined via flow cytometry. (**B**) Percentage of cell populations in G_0_/G_1_, S, and G_2_/M. (**C**) The cycle distribution of HaCaT cells was determined via flow cytometry. (**D**) Percentage of cell populations in G_0_/G_1_, S, and G_2_/M. Data are shown as mean ± S.D. (*n* = 5) for each group. ^##^
*p* < 0.01, ^###^
*p* < 0.001 vs. control group, * *p* < 0.05, ** *p* < 0.01, *** *p* < 0.001 vs. H_2_O_2_−treated group.

**Figure 7 ijms-24-13129-f007:**
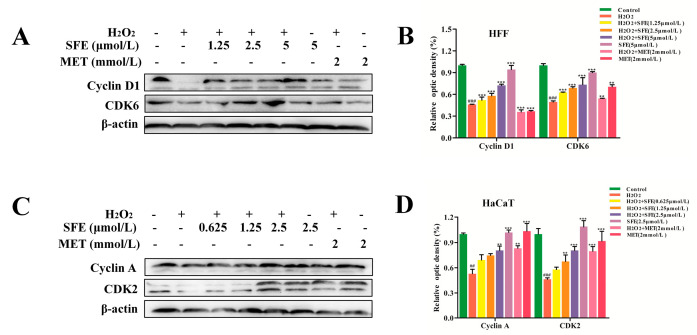
SFE decreased the expression of cell−cycle−arrest−related proteins. (**A**) The expression levels of Cyclin D1 and CDK6 were detected via western blotting analysis. (**B**) Quantification of Cyclin D1 and CDK6 expression. (**C**) The expression levels of Cyclin A and CDK2 were detected via western blotting analysis. (**D**) Quantification of Cyclin A and CDK2 expression. ^##^
*p* < 0.01, ^###^
*p* < 0.001 vs. control group, ** *p* < 0.01, *** *p* < 0.001 vs. H_2_O_2_−treated group.

**Figure 8 ijms-24-13129-f008:**
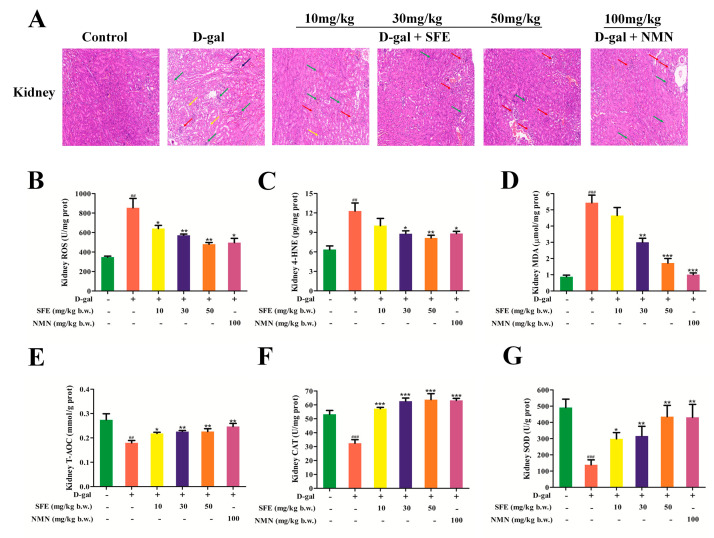
SFE ameliorated D−gal−induced oxidative kidney damage in mice. (**A**) Representative image of the kidney (200× magnification). The red arrows indicate the glomerulus. Inflammatory cell infiltration is indicated by green arrows. Kidney cell vacuolation and degeneration are indicated by black arrows. Kidney cell fibrosis is indicated by the yellow arrows. Kidney ROS level (**B**), 4−HNE level (**C**), MDA level (**D**), T−AOC activity (**E**), CAT activity (**F**), and SOD activity (**G**). Data are expressed as mean ± S.D. (*n* = 6–8). ^##^
*p* < 0.01, ^###^
*p* < 0.001 vs. control group, * *p* < 0.05, ** *p* < 0.01, *** *p* < 0.001 vs. D−gal−treated group.

**Figure 9 ijms-24-13129-f009:**
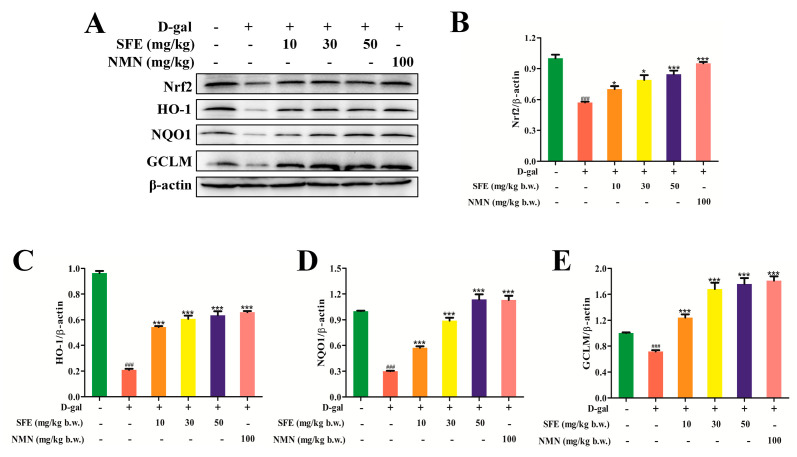
SFE enhanced the antioxidant capacity of the kidney by activating the Nrt2–ARE signaling pathway. (**A**) The expression levels of Nrf2, HO−1, NQO1, and GCLM were detected via western blot analysis. (**B**) Quantification of Nrf2 expression. (**C**) Quantification of HO−1 expression. (**D**) Quantification of NQO1 expression. (**E**) Quantification of GCLM expression. Data are expressed as mean ± S.D. (*n* = 6–8). ^###^
*p* < 0.001 vs. control group, * *p* < 0.05, *** *p* < 0.001 vs. D−gal−treated group.

**Figure 10 ijms-24-13129-f010:**
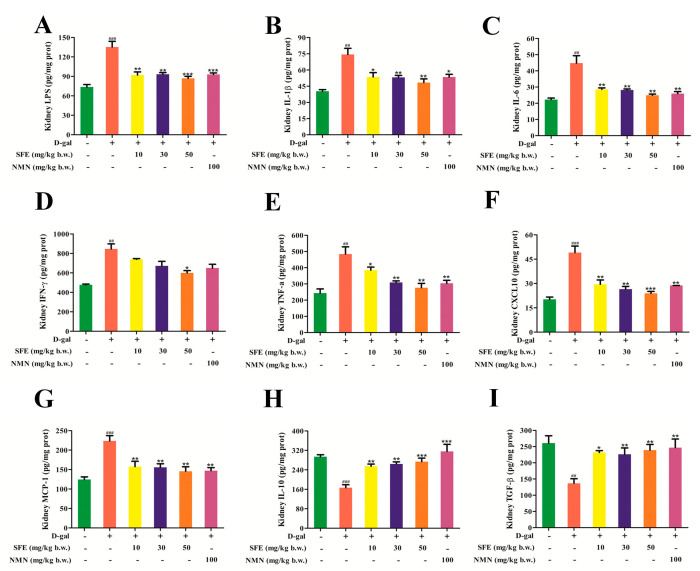
SFE attenuated D-gal-induced inflammatory responses in the kidneys of mice. (**A**) Kidney LPS level. (**B**) Kidney IL-1β level. (**C**) Kidney IL-6 level. (**D**) Kidney IFN-γ level. (**E**) Kidney TNF-α level. (**F**) Kidney CXCL10 level. (**G**) Kidney MCP-1 level. (**H**) Kidney IL-10 level. (**I**) Kidney TGF-β level. Data are expressed as mean ± S.D. (*n* = 6–8). ^##^
*p* < 0.01, ^###^
*p* < 0.001 vs. control group, * *p* < 0.05, ** *p* < 0.01, *** *p* < 0.001 vs. D-gal-treated group.

**Figure 11 ijms-24-13129-f011:**
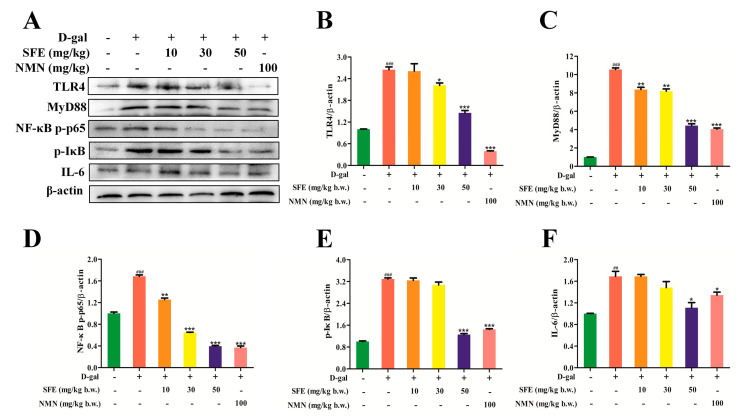
SFE SFE reduced the D-gal-treated mice’s levels of kidney TLR4 and downstream protein expression. (**A**) The expression levels of TLR4, MyD88, NF-κB p-p65, p-IκB, and IL-6 were detected via western blot analysis. (**B**) Quantification of TLR4 expression. (**C**) Quantification of MyD88 expression. (**D**) Quantification of NF-κB p-p65 expression. (**E**) Quantification of p-IκB expression. (**F**) Quantification of IL-6 expression. Data are expressed as mean ± S.D. (*n* = 6–8). ^##^
*p* < 0.01, ^###^
*p* < 0.001 vs. control group, * *p* < 0.05, ** *p* < 0.01, *** *p* < 0.001 vs. D-gal−treated group.

**Table 1 ijms-24-13129-t001:** Kidney index and body weight change in mice.

Groups	Kidney Index (mg/g)	Initial Body Weight (g)	Final Body Weight (g)
Control	14.78 ± 0.79	23.9 ± 0.5	27.8 ± 1.0
D−gal	12.44 ± 0.86 ^##^	22.4 ± 0.8	24.9 ± 1.3 ^###^
D−gal + SFE (10 mg/kg b.w.)	13.41 ± 0.94	23.9 ± 1.0	25.2 ± 0.9 *
D−gal + SFE (30 mg/kg b.w.)	13.51 ± 0.59	23.7 ± 0.6	26.8 ± 0.8 ***
D−gal + SFE (50 mg/kg b.w.)	14.84 ± 1.56 **	22.7 ± 0.4	27.2 ± 1.0 ***
D−gal + NMN (100 mg/kg b.w.)	13.35 ± 0.72	23.4 ± 0.8	26.9 ± 1.2 ***

Data are expressed as mean ± S.D. (*n* = 6–8). ^##^
*p* < 0.01, ^###^
*p* < 0.001 vs. control group, * *p* < 0.05, ** *p* < 0.01, *** *p* < 0.001 vs. D−gal−treated group.

## Data Availability

The data presented in this study are available in https://www.mdpi.com/article/10.3390/ijms241713129/s1.

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
