# Peer review of "New Insight into the Potential Protective Function of Sulforaphene against ROS−Mediated Oxidative Stress Damage In Vitro and In Vivo"

_ijms, 2023, doi:10.3390/ijms241713129_

Round 1

Reviewer 1 Report

The publication describes numerous studies, the results of which indicate the anti-inflammatory properties of sulforaphene. This publication seems to be within the scope of journal. However it needs several corrections to be more acceptable for publication.

1.      The abstract should be rewritten as it must be clear which results are from in vitro reserch and which are from experiments conducted on mice.

2.      Line 13: „in vitro” and „in vivo” should be written in italic.

3.      Line 77 Please add information, that HFF is cell line of fibroblasts and HaCaT is aneuploid keratinocyte cell line from human skin. Please add short explanation of why these specific cell lines were selected for the study.

4.      In materials and methods please add information of what collection the cell lines came from.

5.      It is known that sulforaphene is very unstable in water and other solvents. It is strong electrophilic and it has a tendency to react with the nucleophilic groups of water molecule. See: Zhang, J., Li, X., Ge, P., Zhang, B., Wen, L., Gu, C., Zhou, X. Sulforaphene: Formation, stability, separation, purification, determination and biological activities. Separation & Purification Reviews, 2022, 51(3), 330-339. How the authors ensured the stability of SFE during the studies?

6.      Line 84: It should be „metformin” instead of „Metformin”. In text of manuscript, please add short explanation of why this compound was selected as positive control.

7.      All figures are too small and completety irreadible. E.g. Fig. 1 should be stretched across the entire page. Please arrange all figures in the text after the appropriate section, where the obtained results have been presented, e.g. Fig. 1 should be placed after section 2.1.

8.      Line 142 and others: It should be „in G0/G1 phase” instead of „in G0/G1 phase”. Please check carefully the whole manuscript and correct evident mistake.

9.      Line 250: the authors have written that SFE, which is extracted from the seed of cruciferous vegetables, is the most studied and representative compound of isothiocyanate. However, a much better studied compound is its structural analog, sulforaphane, which is known since 1948. Numerous review articles can be found on the biological activity of sulforaphane, e.g. Mahn, A., Castillo, A. Potential of sulforaphane as a natural immune system enhancer: A review. Molecules, 2021, 26(3), 752; Ruhee, R. T., Suzuki, K. (2020). The integrative role of sulforaphane in preventing inflammation, oxidative stress and fatigue: A review of a potential protective phytochemical. Antioxidants, 9(6), 521; Santín-Márquez, R., Alarcón-Aguilar, A., López-Diazguerrero, N. E., Chondrogianni, N., Königsberg, M. Sulforaphane-role in aging and neurodegeneration. Geroscience, 2019, 41, 655-670.

10.  In the discussion, it would be advisable to discuss the results obtained with those observed for its structural analogue – sulforaphane.

11.  Recent publications on sulforaphene activity should be included in the discussion of results, such as: Yao, H., Du, Y., Jiang, B., Liao, Y., Zhao, Y., Yin, M., ... & Du, M. Sulforaphene suppresses RANKL-induced osteoclastogenesis and LPS-induced bone erosion by activating Nrf2 signaling pathway. Free Radical Biology and Medicine 2023, 207, 48-62; Ye, Q., Yan, T., Shen, J., Shi, X., Luo, F., Ren, Y. Sulforaphene targets NLRP3 inflammasome to suppress M1 polarization of macrophages and inflammatory response in rheumatoid arthritis. Journal of Biochemical and Molecular Toxicology, 2023, e23362.

Reviewer 2 Report

The work is well written, interesting, I recommend to accept.

The article is devoted to the study of the bioactivity of sulforaphene isolated from radish seeds. Previously, for this compound, as well as for the known sulforaphane, therapeutic effects were established. The work investigates the protective effect of sulforaphene in processes caused by oxidative stress. It has been established that sulforaphene mitigates cytotoxicity, exhibits protective effects by improving mitochondrial function, reducing apoptosis and accelerating the progression of the cell cycle. Suppression of excessive production of ROS contributes to the protection of the kidneys from oxidative damage. The effectiveness of the protective action of the studied natural compound allows the authors to conclude that sulforaphen can be used not only as a prophylactic agent for kidney damage, but also as a promising rejuvenating agent. The high interest of researchers at present in natural plant products that can act as effective and safe medicines determines the relevance of the presented work. In general, the work is well written, all comments are limited to correcting typos and minor errors that can be corrected at the stage of proof. As a wish, we can add that it would be nice to improve the quality of the drawings, since some of them are hard to read.

Reviewer 3 Report

The manuscript “New insight into the potential protective function of sulforaphene against ROS-mediated oxidative stress damage in vitro and in vivo “ fits the journal’s scope. The authors present their findings regarding the effect of sulforaphene on in vitro and in vivo models of oxidative stress damage. The aim of the study is clearly stated. Overall, the methods are well presented, but the references are missing in all subsections of material and methods. The in vivo methodology should be updated, and more details should be provided. The results are presented in sufficient detail, but the figures should be inserted in their places. The discussions are well presented, and the conclusion is sustained by the authors’ findings. Before publication, the authors should correct the manuscript.

The figures should be inserted at their place in the manuscript.

Lines 68-72 – please indicate the cell lines

Lines 29, 76, 252, 342-343, 338-340 - please correct/rephrase

Lines 75-76, 106-108, 203-214, 223-224,– please add references

The references are missing in all sections of material and methods. Please add them.

Table 1 – please explain the importance of parameter “body weight”

Section 5.8 – please describe in more detail

Section 5.9

-        Although the design of the experiment is partially explain in section 2.5, a brief description should be add in this section

-        (also lines 518-520) please indicate the guidelines that applied to this research; please provide the approval letter of the Bioethical Committee (of the University, and not the letter from the supplier), or fully justify if this does not exist; please add references of the guidelines followed.

-        please add information regarding the mice euthanasia method and conditions ; if OECD guidelines were followed, please provide the necropsy’ results or add justifications (if it wasn’t performed)

5.10 – please add the missing references

5.13 -  please indicate the number of replicates

Please improve the quality of figures (including all graphs)

Fihure 7 – please indicate the damage induced by D –gal (preferably using arrows)

Please use subscript for H2O2

Round 2

Reviewer 3 Report

The authors addressed the raised issues and made the corrections. In the present form, the manuscript is suitable for publication.

Please see minor corrections:

Line 508 – please correct the errors
